# Mechanism for nuclease regulation in RecBCD

**Martin Wilkinson, Yuriy Chaban, Dale B Wigley***

Section of Structural Biology, Department of Medicine, Imperial College London, London, United Kingdom

**Abstract** In bacterial cells, processing of double-stranded DNA breaks for repair by homologous recombination is catalysed by AddAB, AdnAB or RecBCD-type helicase-nucleases. These enzyme complexes are highly processive, duplex unwinding and degrading machines that require tight regulation. Here, we report the structure of *E.coli* RecBCD, determined by cryoEM at 3.8 Å resolution, with a DNA substrate that reveals how the nuclease activity of the complex is activated once unwinding progresses. Extension of the 5'-tail of the unwound duplex induces a large conformational change in the RecD subunit, that is transferred through the RecC subunit to activate the nuclease domain of the RecB subunit. The process involves a SH3 domain that binds to a region of the RecB subunit in a binding mode that is distinct from others observed previously in SH3 domains and, to our knowledge, this is the first example of peptide-binding of an SH3 domain in a bacterial system.

## Introduction

The repair of double-strand DNA breaks is a critical process in all organisms. In bacteria, one of the major pathways for high fidelity repair of breaks is homologous recombination initiated by RecBCD, AdnAB or AddAB complexes (*Dillingham and Kowalczykowski, 2008*; *Wigley, 2013*) that process the broken ends via a number of steps to eventually produce a RecA-coated 3'-tailed filament that initiates strand invasion.

In addition to a role in repair of double-strand DNA breaks, RecBCD is an important player in controlling infection by bacteriophages, the significance of which is exemplified by proteins produced by phages that specifically target RecBCD. The phage lambda protein Gam, works by competing with DNA binding to RecBCD (*Murphy, 2007*; *Court et al., 2007*), while the phage P22 protein Abc2, works by inducing a 'Chi-like state' for RecBCD (*Murphy, 2000*). Most recently, a role for RecBCD in providing DNA fragments for CRISPR arrays demonstrates a connection to the other major pathway known to provide resistance to phage infections (*Levy et al., 2015*).

The RecBCD/AddAB complexes are highly processive helicase/nuclease machines that digest DNA from the broken end until they encounter a crossover hotspot instigator (Chi) sequence. At this point their activities are modulated to produce a 3-tailed duplex onto which RecA is loaded (*Wigley, 2013*). Crystal structures of a variety of intermediates on the AddAB-mediated pathway (*Saikrishnan et al., 2012*; *Krajewski et al., 2014*), together with biochemical and single molecule experiments (*Gilhooley and Dillingham, 2014*; *Gilhooley et al., 2016*; *Carrasco et al., 2013*), now provide a reasonable understanding of the molecular details of this process.

Although there has been considerable study of RecBCD at the biochemical and single molecule level (*Wright et al., 1971*; *Taylor and Smith, 1992*, *2003*; *Dillingham and Kowalczykowski, 2008*; *Spies et al., 2003*, *2007*), our understanding of the molecular details for the RecBCD mechanism are less advanced due to a lack of high resolution structural information. To date, this has been limited to the initiation complex, with the protein interacting with a broken DNA end (*Singleton et al.,*

*For correspondence: d.wigley@imperial.ac.uk

**Competing interests:** The authors declare that no competing interests exist.

*2004*), and some DNA substrates representing partially unwound substrates that represent the very initial stages of the pathway (*Saikrishnan et al., 2008*). The nuclease activity of RecBC is very much lower than that of RecBCD (*Anderson et al., 1997*) so it was initially presumed that the nuclease must be a part of the RecD subunit. However, it was shown instead that the nuclease domain is actually a C-terminal domain of the RecB subunit, the activity of which is regulated by RecD (*Yu et al., 1998*). The structure of the initiation complex revealed this is due to an α-helix that sits in the nuclease active site, thus blocking access to DNA (*Singleton et al., 2004*). It is evident that this helix must be displaced to activate the nuclease, but the mechanism and trigger for this release was unknown.

In order to understand how the RecD subunit is able to activate the nuclease domain of RecB, we have now determined the cryo-electron microscopy (cryoEM) structure of RecBCD bound to a forked DNA substrate with an extended 5'-tail that spans the entire RecD subunit. This interaction induces a series of conformational changes that eventually result in release of the nuclease-blocking linker helix, which then re-docks at an alternative location thus opening the nuclease active site for access by the ssDNA tails of the unwound fork. A critical role for a rare bacterial SH3 domain in the RecD subunit is also revealed in the structure, and raises possibilities for a mechanism of further regulation after recognition of Chi.

## Results and discussion

### Structure determination by cryoEM

Fork DNA substrates containing ten residues on the 5'-tail could be crystallised in the same crystal form as the blunt hairpin substrate used for the initiation complex reported previously (*Singleton et al., 2004*; *Saikrishnan et al., 2008*), and the structures of the protein components were essentially the same. However, just a single base addition on the 5'-tail beyond ten residues resulted in loss of crystallisation and we were unable to obtain suitable crystals despite extensive screening. This was the first indication that the additional base on the 5'-tail induces a conformational change in the protein. Consequently, despite the relatively small size of the RecBCD/DNA complex (350KDa), we turned instead to cryoEM to determine a high resolution structure of the complex. Data were collected on a Titan Krios microscope with Gatan K2 detector at the UK EM facility at Diamond. EM processing statistics are presented in the Materials and methods section, *Table 1*, *Figure 1* and *Figure 1—figure supplement 1*. Electron density corresponding to the most important areas described in this manuscript is shown in *Figure 1—figure supplement 2*. Data processing revealed a major class that constituted about half of the particles and this is what is described in detail below. We utilised a DNA fork substrate with a 25 base pair duplex region, capped at one end with a five base hairpin and at the other with unpaired 3' and 5'-tails (3 and 12 thymidine residues, respectively). The DNA and all domains of the three proteins were accounted for in the reconstruction derived from this major class (*Figure 1*). In one of the other classes, density corresponding to the RecB nuclease domain was missing (18%), although other features for the rest of the protein were very similar to the main class. This class may constitute some proportion of the complex in which the flexible linker connecting the nuclease domain to the helicase domains of the RecB subunit has been cleaved or, alternatively, may be due to release of the nuclease domain from its usual binding surface observed in the initiation complex. In the latter instance, we do not see any evidence for a potential docking site for the nuclease domain anywhere else on the complex such as that proposed from SAXS studies (*Taylor et al., 2014*). This second class was therefore of limited interest

**Table 1.** Refinement of the final model.

| | |
|---|---|
| Final particles | 74656 |
| Resolution (Å) | 3.8 |
| CC (whole map) | 0.838 |
| CC (around atoms) | 0.875 |
| R.m.s. deviations | |
| Bond lengths (Å) | 0.01 |
| Bond angles (°) | 1.04 |
| Ramachandran | |
| Favoured (%) | 94.1 |
| Outliers | 8/2874 |
| Rotamers | |
| Favoured (%) | 99.1 |
| Outliers | 1/2426 |

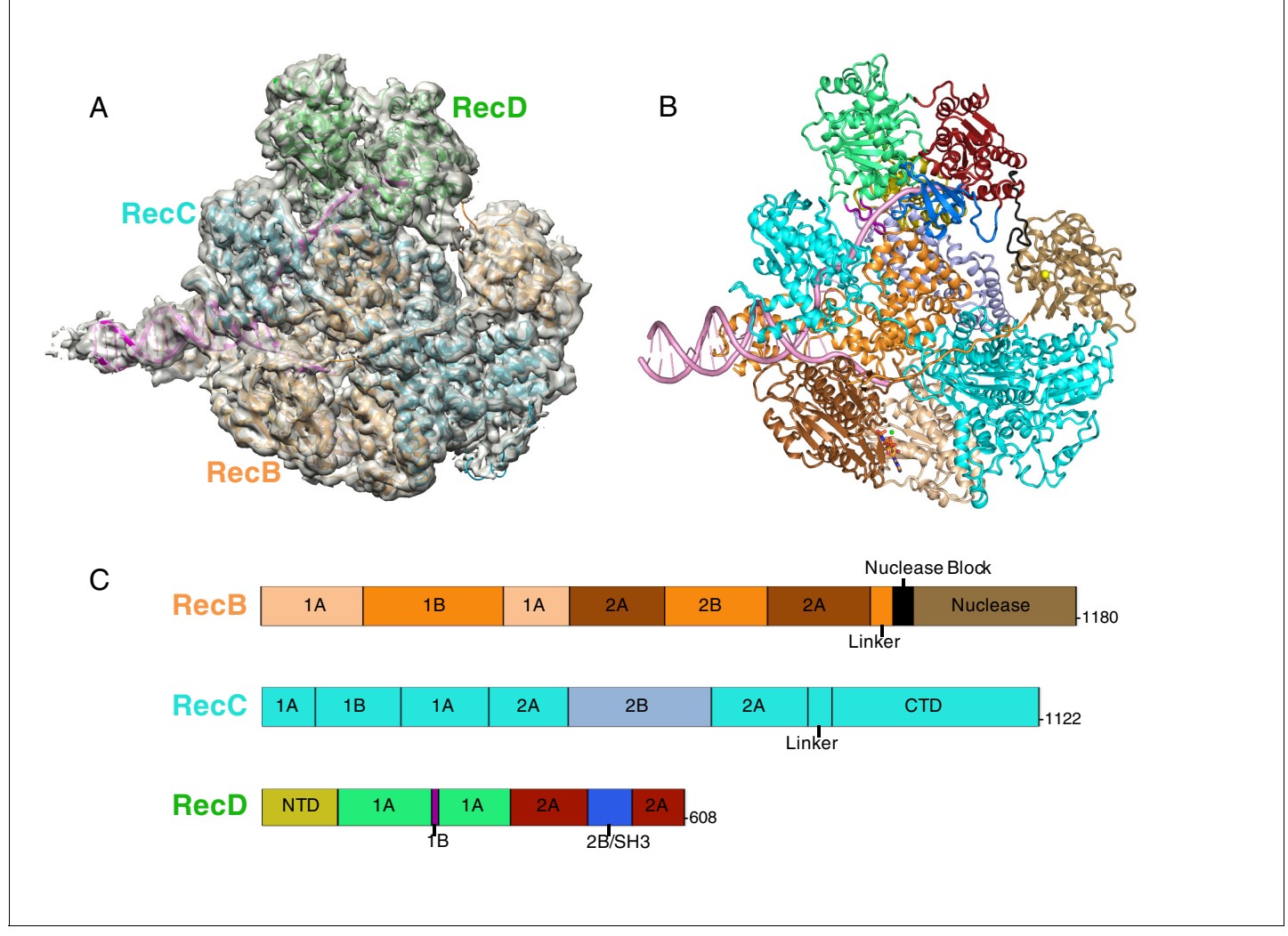

**Figure 1.** CryoEM structure of RecBCD/DNA complex at 3.8 Å resolution. (**A**) Cartoon representation of the RecBCD complex within a transparent surface showing the electron density. The RecB subunit is in orange, RecC in cyan, RecD is in green, and the DNA is in pink, (**B**) 'Road map' for colour scheme used in subsequent figures using the colour scheme shown in (**C**), (**C**) Linear representation of the sequences, with labelling of relevant domains, of the three subunits showing the colour scheme used in subsequent figures. Further information about the quality of the CryoEM structure is presented in *Figure 1—figure supplement 1*. Examples of the quality of the electron density are shown in *Figure 1—figure supplement 2*.

The following figure supplements are available for figure 1:

**Figure supplement 1.** EM Data statistics.

**Figure supplement 2.** Representative sections of electron density (contorted at 4 σ).

so we confine the discussion of the results to the dominant class of particles that appears to represent an intermediate on the catalytic pathway.

## The RecB subunit

Electron density for bound ADPNP is visible in the ATP-binding site of the RecB subunit (*Figure 2—figure supplement 1*). The adenine ring is sandwiched between two aromatic residues (F31 and W447), but otherwise the contacts with active site residues are very similar to those observed in other Superfamily 1 (SF1) helicases (*Singleton et al., 2007*). In response to nucleotide binding, the helicase domains undergo a conformational change (*Figure 2—figure supplement 1*) as expected

from studies of a variety of other SF1 helicases including PcrA and AddAB (*Singleton et al., 2007*; *Velankar et al., 1999*; *Krajewski et al., 2014*). The short 3'-tail we used for this structure (to permit extension of the duplex and 5'-tail) interacts with the 2A domain of the RecB subunit as seen in the initiation complex (*Singleton et al., 2004*). Although RecBCD is able to unwind up to six base pairs from a blunt end in the absence of ATP (*Singleton et al., 2004*; *Wong et al., 2005*), it was unable to further unwind the fork substrate used here.

As seen in previous crystal structures, the 'arm' domain of the RecB subunit extends from the protein core and contacts the incoming duplex across the minor groove (Supplementary *Figure 2—figure supplement 1*; *Singleton et al., 2004*; *Saikrishnan et al., 2008*). This contrasts with the duplex: arm contacts in AddAB which are across the major groove (*Figure 2—figure supplement 1*) (*Saikrishnan et al., 2012*). However, conformational changes observed when nucleotide binds to the RecB subunit (*Figure 2—figure supplement 1*) are very similar to those seen in the AddA subunit within AddAB, suggesting a similar mechanism for unwinding and translocation once the arm has engaged the DNA substrate (*Krajewski et al., 2014*).

## Interactions with the 5'-tail and ADPNP within the RecD subunit

Electron density corresponding to all twelve bases of the 5'-tail could be traced from the fork junction, across and through the RecC subunit, and along a groove on the RecD subunit (*Figure 2*). The first nine bases were essentially the same as the first nine (of ten) in the extended fork complexes (*Saikrishnan et al., 2008*). In addition to these contacts, we were able to pick out the remaining three bases that extended across the 2A domain of RecD (*Figure 2*). The ssDNA bound across the RecD subunit is very similar to that observed in the RecD2 complex and the contacts also involve similar conserved residues (*Saikrishnan et al., 2009*).

Surprisingly, only very weak electron density for bound nucleotide is visible at the ATP-binding site of RecD even though ADPNP is bound at the ATP-binding site of the RecB subunit (see *Figure 2—figure supplement 2* and discussion in the figure legend).

## Structure of the SH3-fold RecD 2B domain

The 2B domain of the RecD subunit was disordered in the crystal structure of the initiation complex reported previously (*Singleton et al., 2004*). However, the crystal structure of the related *D. radiodurans* RecD2 protein (*Saikrishnan et al., 2009*), and more recently of the related SF1B family helicases Dda (*He et al., 2012*) and Pif1 (*Zhou et al., 2016*; *Chen et al., 2016*), revealed an SH3 fold for this domain, which is rare in prokaryotic systems. These structures also revealed that the SH3 domain interacts with the ssDNA tail in a location different to that normally occupied by a peptide in canonical eukaryotic SH3 domains (*Saikrishnan et al., 2009*; *Saksela and Permi, 2012*), thus retaining the potential to bind peptide at the same time as the ssDNA tail. The 2B domain of the RecD subunit is visible in our current EM structure and confirms the fold of this domain is similar to that of the SH3 fold observed for RecD2. It also confirms the location and mode of DNA binding observed in RecD2. However, the domain also contacts a helix within the 2B domain of RecB (*Figure 3A*). This interaction utilises a surface that is in a similar location to that used by other SH3 domains (*Saksela and Permi, 2012*) but the mode of interaction is very different to other SH3 domains. Most SH3 domains bind an extended peptide (usually a sequence rich in proline residues) across a groove on the protein surface. By contrast, the SH3 domain of RecD binds an α-helix in this region. In addition, some of the regions that constitute the contact surfaces in canonical eukaryotic SH3 domains, such as in Abl kinase, are utilised in a different manner in the RecD SH3 domain (*Figure 3B*). The RT and N-Src loops are pulled back from their usual locations through interactions with the ssDNA tail and a part of the RecD 2A domain, respectively. This opens up the typical narrow peptide-binding groove that opens up the surface to interact with an α-helix in the RecB subunit. Therefore, this appears to represent a novel interaction mode for a SH3 domain.

Finally, a short extension in the region immediately preceding the SH3 domain also binds a polypeptide linker that connects the RecB helicase and nuclease domains (see below). Consequently, this small domain, that is disordered in the initiation complex, becomes the focal point of three separate interaction interfaces that together coordinate a series of conformational changes during nuclease activation (see below).

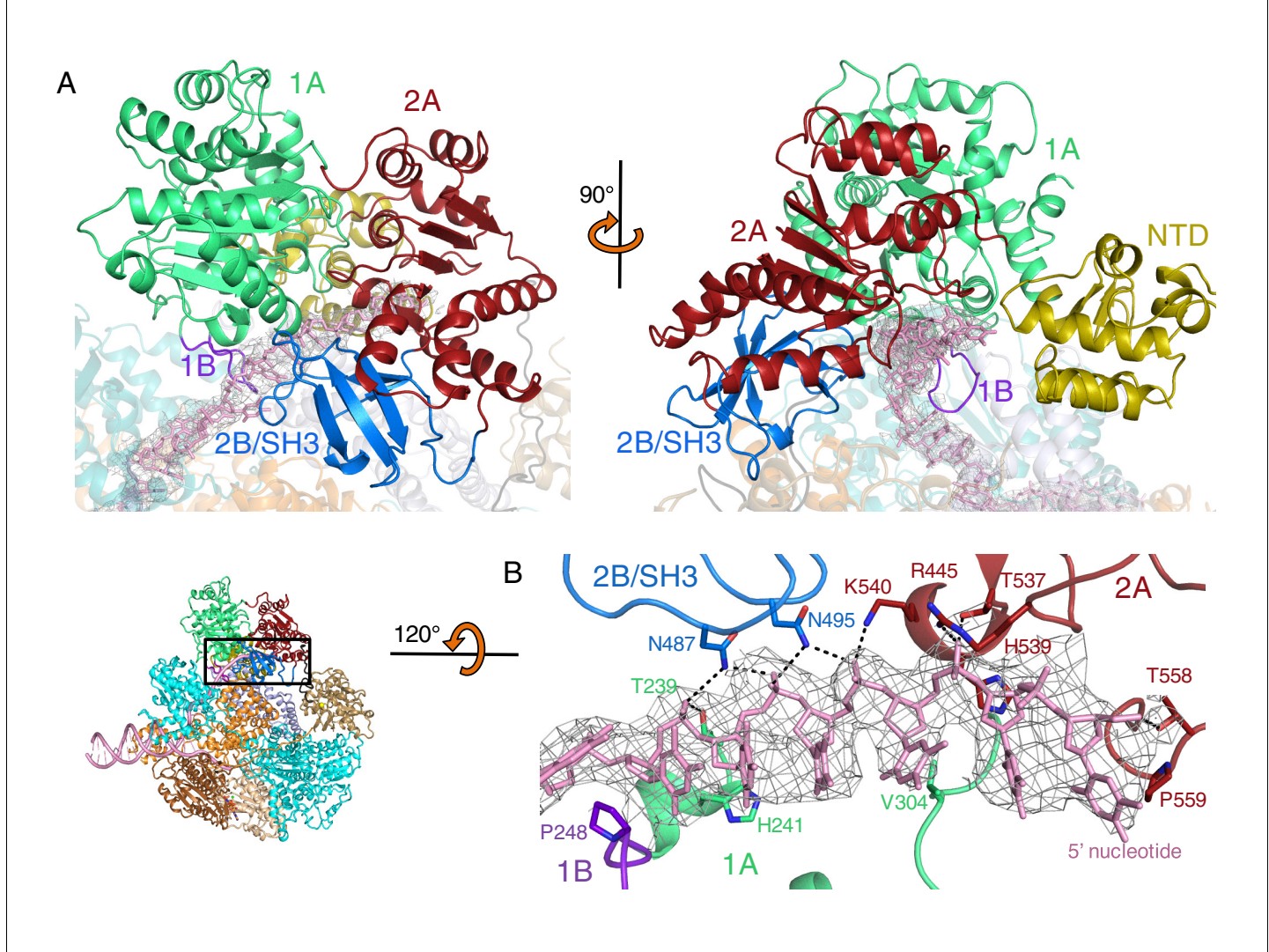

**Figure 2.** Interactions with the 5'-tail and the RecD subunit. Inset: location of the close ups shown in this figure relative to *Figure 1*. (**A**) Binding of the 5'-tail of the DNA fork across the RecD subunit, (**B**) Close up of the electron density (contoured at 4 σ) for the bound 5'-tail of the DNA fork showing details of the contacts between the protein and bound DNA. Further details about interactions within the RecD subunit are presented in *Figure 1—figure supplement 1*, and details about interactions within the RecB subunit are presented in *Figure 2—figure supplement 2*

The following figure supplements are available for figure 2:

**Figure supplement 1.** Structural features of the RecB subunit.

**Figure supplement 2.** Nucleotide binding at the RecD subunit.

Genetic and biochemical data show that the RecD subunit is not required after Chi recognition (*Stahl et al., 1990*; *Dixon et al., 1994*). This led to the so-called RecD ejection hypothesis (*Stahl et al., 1990*). However, subsequent single molecule data have shown instead that RecD becomes inactivated rather than being lost from the complex (*Handa et al., 2005*). If it were to act in a conventional SH3 mode (i.e. canonical peptide binding) then it is plausible that conformational changes in response to Chi recognition might cause this domain to relocate and bind to another part of the protein complex to lock RecD in a conformation that can no longer function as a helicase/translocase, without being lost from the complex. This could either involve a similar region to that we observe in the current structure or a different one following another conformational change.

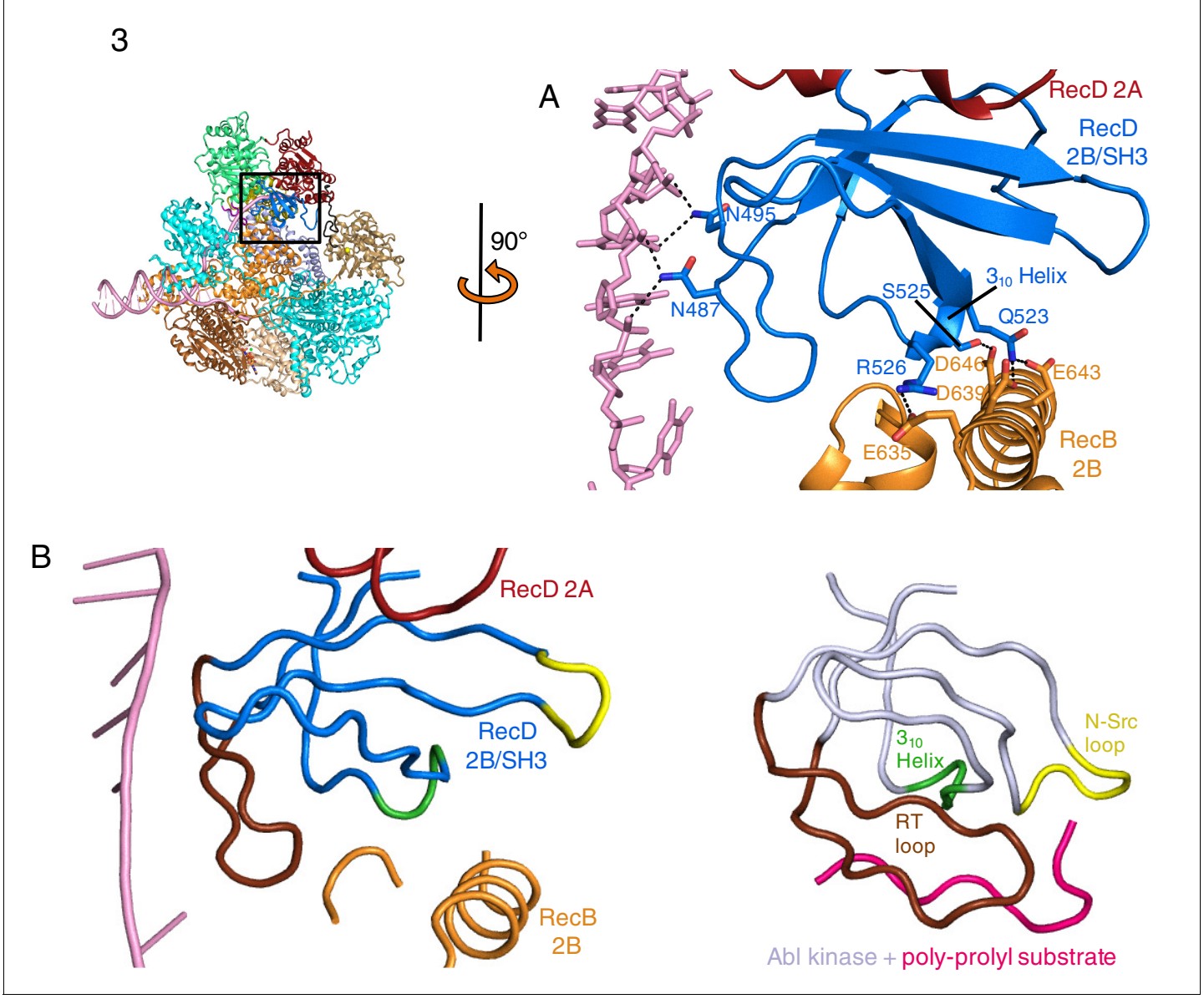

**Figure 3.** SH3 fold in RecD. Inset: location of the close ups shown in this figure relative to *Figure 1*. (**A**) Contact regions of the SH3 domain (blue) with DNA (pink) and peptide (RecB, orange), (**B**) Cartoon representation of the SH3 domain of Abl kinase (right) with bound peptide (magenta). The key contact regions (*Saksela and Permi, 2012*) are highlighted in different colours. The equivalent view of the RecD SH3 domain is shown on the left.

## Conformational changes in response to extension of the 5'-tail

In previous structures of RecBCD, the RecD subunit adopted an open conformation but in the current structure the protein closes around and clamps the 5' ssDNA tail. The closing of the RecD clamp brings the SH3 domain of RecD into contact with the 2B domain of the RecB subunit that appears to act as a catch to stabilise the closed conformation of RecD (*Figure 4*, *Figure 4—figure supplement 1*, and *Videos 1–3*). The N-terminal domain of RecD contacts the 2B domain of RecC and the conformational change in RecD has significant effects on the conformation of the RecC subunit. The RecB nuclease domain bridges a gap between the 1A and 2B domains of RecC and the conformational changes cause a significant shift in position of the RecB nuclease relative to these domains that reveals the mechanism for nuclease activation.

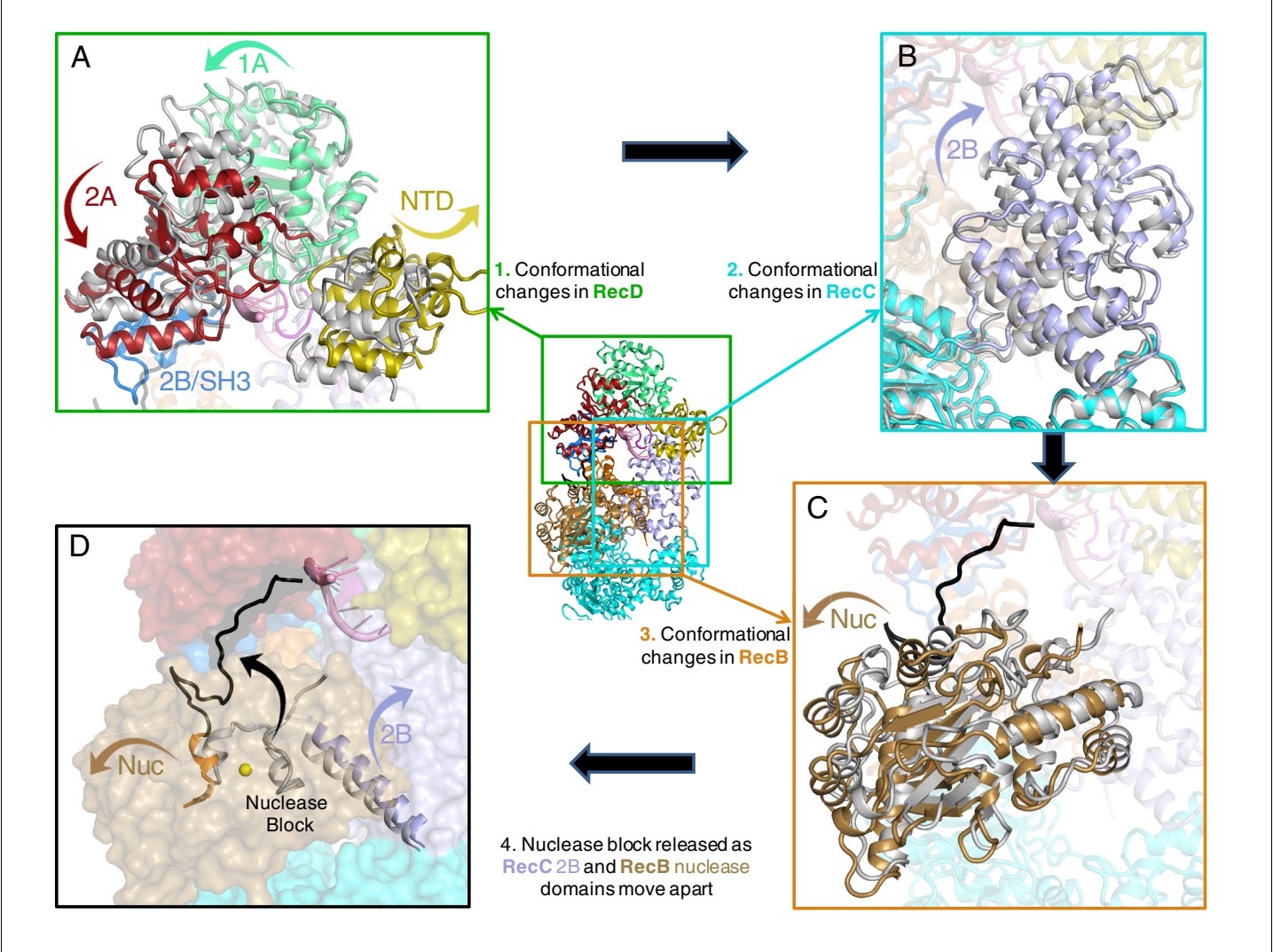

**Figure 4.** Conformational changes that result in nuclease activation. Inset: location of the close ups shown in this figure. The series of conformational changes from the initiation complex (grey) begins in the top left corner. Supplementary *Videos 1–3* show these conformational changes. (A) Overall changes in the structure in the RecD subunit. Note that the SH3 domain is disordered in the initiation complex (see also *Videos 1* and *2*), (B) Overall changes in the structure in the RecC subunit (see also *Video 2*), (C) Overall changes in the structure in the RecB nuclease domain (see also *Video 2*), (D) The linker polypeptide region, including the α-helix (black) that blocks the active site, relocates to a new position (see also *Video 3*). Electron density for this region is shown in *Figure 4—figure supplement 1*.

The following figure supplement is available for figure 4:

**Figure supplement 1.** Electron density showing relocation of the linker peptide and nuclease blocking helix.

## Mechanism of nuclease activation

The nuclease activity of the RecBCD complex is attenuated in the initiation complex prior to binding ATP (*Wright et al., 1971*). The crystal structure of the initiation complex of RecBCD suggested why this is the case because the nuclease requires ssDNA to be fed into it by the helicase activities of the complex. Furthermore, the structure revealed a mechanism for inhibition of the nuclease activity in which an α-helix (residues 913–922) within the linker region (residues 870–940) of RecB, that connects the C-terminal nuclease domain to the N-terminal helicase domains, sits in the nuclease active site thus blocking access (*Singleton et al., 2004*). This suggested an obvious mechanism for activation in which this helix would be displaced to allow access of the ssDNA tails to the nuclease site

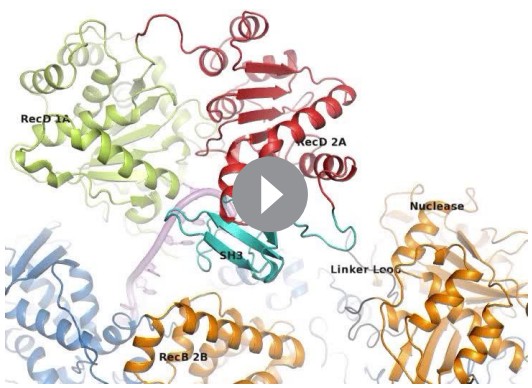

**Video 1.** Overall changes in the RecD subunit.

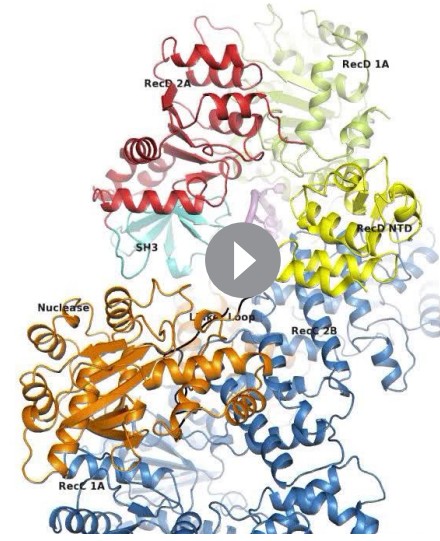

**Video 2.** Overall changes in the RecC subunit.

although how this might be brought about was not evident from the structure. The EM structure now reveals the details of this process (*Figure 4* and *Videos 1–3*).

Interaction between the 2B domain of RecD and the RecB subunit has a number of consequences. First, several domains of the RecC subunit also undergo a change in conformation. This change alters the interaction between the 1A domain of RecC and the nuclease domain of RecB. In the initiation complex, the α-helix that blocks the nuclease active site is stabilised by stacking contacts with another α-helix (residues 602–627) from the 2B domain of RecC. This interaction is altered as a consequence of the conformational changes in the RecC subunit that shift the position of the RecC helix, breaking the contacts with the nuclease-blocking helix of the RecB linker, resulting in release of this helix from the nuclease active site, for which there is now no electron density (*Figure 4—figure supplement 1*). This change opens up the nuclease site and allows access of DNA. Furthermore, weak electron density suggested an alternate path for the linker region compared to the crystal structure (*Figure 4—figure supplement 1*). Although the density was too weak to build the loop accurately, it was modelled as poly-alanine to define the overall pathway for this re-docked polypeptide. The path takes the linker along the edge of the RecD 2A domain including a short region that precedes the SH3 domain. Thus the entire linker is relocated away from the nuclease domain, relieving the block and providing access to the nuclease active site. The conformational changes are illustrated by movies created by morphing from one structure to the other (*Videos 1–3*). A cartoon representing the steps on the RecBCD activation pathway is shown in *Figure 5*.

The activity of a highly processive helicase/nuclease could be very deleterious to a cell if unchecked. Consequently, RecBCD has evolved regulatory mechanisms that suppress activity unless the enzyme complex is bound to a *bona fide* DNA substrate (i.e. a broken DNA end). The structure we present here demonstrates the molecular basis for one such regulatory mechanism and, together with our previous structural studies, describes a pathway for this activation process. Initial binding of RecBCD to a broken end induces unwinding of up to six base pairs of DNA in the absence of ATP (*Singleton et al., 2004*, *Wong et al., 2005*). The function of this unwinding is to fill up the ssDNA-binding site on the RecB subunit. ATP binding and hydrolysis now allow unwinding of the DNA duplex to

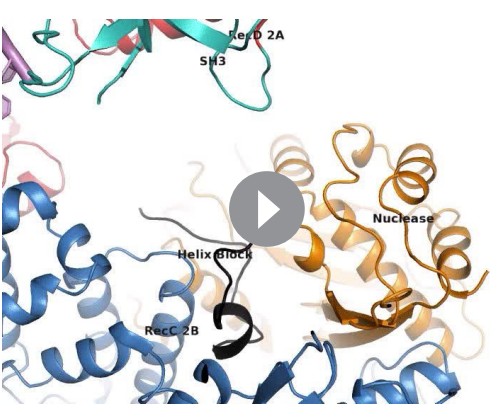

**Video 3.** Changes in the linker peptide region.

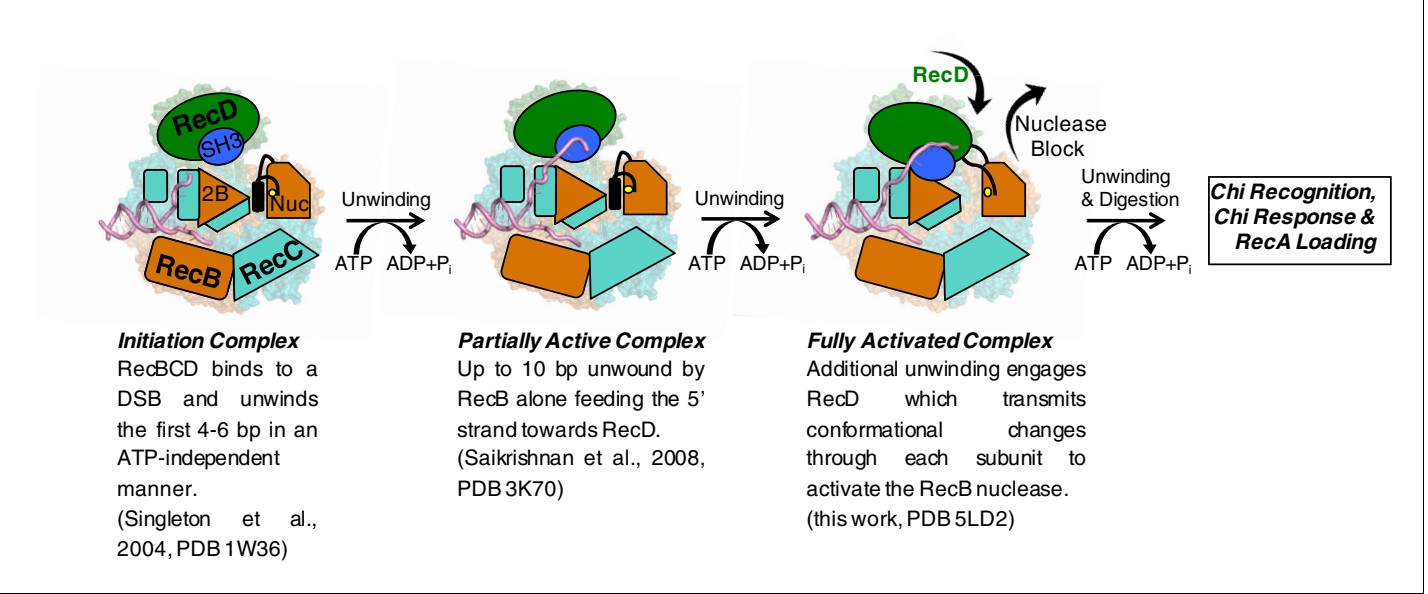

**Figure 5.** Cartoon summarising the conformational changes that proceed from the initiation complex to nuclease activation.

begin, at first driven by just the RecB subunit. At this stage the nuclease active site is blocked by the linker helix and activity of the domain is attenuated (*Saikrishnan et al., 2008*). As the RecB subunit begins to unwind the duplex, the 5'-ssDNA tail that is produced passes along a channel through the RecC subunit towards the motor domains of the RecD subunit. Once the tail exceeds ten bases, the RecD subunit responds with a large conformational change that converts it to an active helicase/ translocase conformation and in this process also transmits this change to the nuclease active site via the RecC and RecB subunits. At this point the enzyme is bound to a DNA substrate that it recognises as being initiated from a broken DNA end (or restriction enzyme digested phage genome) so can proceed to processively and rapidly digest the DNA and search for a Chi sequence. If the enzyme encounters a Chi sequence then it pauses and alters its activities to prepare the DNA end for RecA loading. RecBCD then loads the RecA protein so that homologous recombination can initiate and the repair process proceeds. For phage DNA that lacks Chi sequences, the DNA degrading machine is unregulated and continues to digest the phage genome, thereby eliminating the phage from the cell, but also at the same time providing a molecular memory of the infection via the CRISPR system to further evade future infections (*Levy et al., 2015*).

## Materials and methods

### Protein and DNA preparation

A nuclease-dead *E.coli* RecBCD mutant (D1080A in RecB) was used in this work to prevent digestion of the single-stranded tails of the DNA substrate. The complex was produced from three plasmids, pETduet-His$_6$-TEVsite-recB$^{D1080A}$, pRSFduet-recC and pCDFduet-recD in a ΔrecBD *E.coli* strain as described previously (*Saikrishnan et al., 2008*). Purification of RecBCD was as described previously (*Singleton et al., 2004*) but modified to include a HisTrap Ni-Sepharose (GE Life Sciences) step after ammonium sulphate precipitation. The HisTrap elute was further purified using HiTrap heparin-Sepharose (GE) then dialysed overnight with TEV protease to remove the His-tag. The sample was then passed back through HisTrap Ni-Sepharose before continuing with MonoQ (GE) anion exchange. The protein was flash-frozen after supplemention with 15% glycerol. The hairpin DNA substrate that was used had the sequence 5'- TTTTTTTTTTTTtctaatgcgagcactgctacagcatTTCC-CatgctgtagcagtgctcgcattagaTTT-3', where lower case denotes paired bases in the hairpin stem, with twelve unpaired thymidine nucleotides on the 5' end and three on the 3' end. The substrate was prepared as described previously (*Singleton et al., 2004*; *Saikrishnan et al., 2008*).

## Cryo-electron microscopy grid preparation and data collection

Prior to preparing grids, RecBCD was thawed and desalted into 20 mM Tris-HCl pH 7.5, 50 mM NaCl, 1 mM TCEP using Sephadex G25 spintrap columns (GE) and concentrated to 5 mg/ml. The protein was mixed with a 1.75 fold excess of DNA substrate for 20 min at room temperature before 2 mM ADPNP and 4 mM MgCl$_2$ were added and the complex placed on ice for immediate use.

Samples for cryo-EM were frozen using C-flat advanced holey carbon film grids (Protochips). Grids with 1 and 2 μM round holes were used. In order to obtain larger areas of thin ice inside the holes, the carbon film was thinned by subjecting grids to 2–3 min of glow discharge (10 mA, 0.2 mbar). Thinned grids were kept for at least two weeks prior to use to allow charge dissipation. To make the carbon surface hydrophilic, prior to the sample preparation, grids were treated with the detergent n-Dodecyl β-D-maltoside (DDM) (Sigma-Aldrich) using a procedure described in (*Cheung et al., 2013*) with some modifications. Grids were incubated on 30 μL drops of 0.1% solution of DDM overnight (or for at least 4 hr) at 4°C. Grids were edge-blotted and washed eight times in 10 μL drops of water (the tips of the tweezers for grid handling were wiped each time to improve reproducibility). Washed grids were kept on damp filter paper and used for freezing within two hours of preparation. Sample (3 μL) was applied to the grid and evenly spread with the pipet tip before blotting and freezing in liquid ethane (Vitrobot Mark III (FEI)). This grid treatment allowed us to obtain a significant area of thin ice to provide sufficient numbers of the thinner 'flat' views of the particles, which were harder to find in thicker ice on glow discharged grids. Similar results were achieved by treating grids with a 1 mM solution of Amphipol A8-35 (Anatrace, Maumee, Ohio). Five washes were required to ensure removal of unbound amphipol while providing sufficient grid hydrophilicity.

Data were collected using a Titan Krios microscope operated at 300 KV at eBIC, Diamond, U.K. Zero loss energy images were collected automatically using the EPU software (FEI) on a Gatan K2-Summit detector in counting mode with a pixel size corresponding to 1.34 Å at the specimen. A total of 1674 micrographs were collected in movie mode with a nominal defocus range of −1.2 to −3.2 μm. Each movie consisted of a stack of 30 frames, collected with a total dose of 36 electrons/Å$^2$ over 12 s corresponding to a dose rate of 5.4 electrons/pixel/s.

## Data processing

All 30 movie frames for each image stack were aligned and summed using Motioncorr (*Li et al., 2013*). Subsequent image processing used Relion1.4 (*Scheres, 2012*) unless stated otherwise. The actual defocus and other contrast transfer function (CTF) parameters for each summed movie stack were estimated using CTFFIND4 (*Rohou and Grigorieff, 2015*). Outlying micrographs were removed, based on a number of criteria, leaving 1426 images with an estimated resolution range from 7.9–2.8 Å (mean of 3.9 Å) and defocus range between −0.5 to −3.8 μm. Manually picked particles from a small subset of micrographs were used to generate initial 2D classes which were used as references for autopicking. Initially, reference-free 2D classification was used to remove poor particles from autopicking, leaving 153,334 particles judged to represent RecBCD complexes. These were subjected to 3D classification using the RecBCD:DNA crystal structure (*Singleton et al., 2004*) (PDB 1W36), low-pass filtered to 50 Å, as a starting model. From five initial classes of particles, two appeared identical and contained the full RecBCD:DNA complex (combined total, 49% of dataset) so were pooled and selected for further refinement. Of the remaining classes, one contained RecBCD:DNA apparently lacking the entire RecB nuclease domain (18%), one contained a mixture of molecular species (25%), and one was featureless, containing poorly aligned particles (8%). Particles from these latter three classes were discarded.

3D refinement of the remaining 74,656 particles, using the same starting model as the 3D classification step, yielded a map with a resolution of 4.6 Å before masking (as estimated by the 'gold-standard' Fourier Shell Correlation (FSC) at the 0.143 cut-off criterion). After application of an auto-generated mask in Relion, the resolution of the map was 4.0 Å. Further filtering of the particle dataset by a variety of selection criteria did not improve the final map. Movie processing, using particle polishing to correct for individual particle movement and apply per-frame weighting before re-running 3D refinement, improved the resolution to 3.9 Å. A final resolution of 3.8 Å was obtained by re-refining the 3D model after local CTF parameters for the particles were determined using the

program Gctf (*Zhang, 2016*). The final applied b-factor for sharpening used in post-processing was $-60$ Å$^2$. The final map was deposited at the EM databank with EMD code 4038.

## Model building and refinement

Scripts for map conversion, cell matching and refinement in Refmac were kindly provided by Garib Murshudov (MRC-LMB). The crystal structure of RecBCD in complex with an extended DNA fork (*Saikrishnan et al., 2008*) (PDB 3K70), was used as a template for building in Coot (*Emsley et al., 2010*) after initial global docking in chimera (*Petterson et al., 2004*). There were significant conformational changes apparent, so the jiggle-fit and morph, with positional restraints, tools in Coot were used to obtain a better starting fit. The SH3 domain of RecD, that was absent from the crystal structure, was built manually from a homology model generated using the Phyre2 server (*Kelley et al., 2015*) and based on the structure of *Deinococcus radiodurans* RecD2 (*Saikrishnan et al., 2008*). One molecule of ADPNP and a magnesium atom were modelled into strong density in the RecB helicase domains and positionally-restrained based on bond distances from similar helicases such as in the AddAB crystal structures (*Krajewski et al., 2014*). There was insufficient density to confidently place a molecule of ADPNP in the RecD helicase domains. Weak electron density suggested an alternate path for residues 913–938 in RecB compared to the crystal structure. The density was too weak to accurately build the loop and was modelled in as poly-alanine, with occupancy set to 0 and with atom B-factors set to 100. The entire model was carefully edited and fitted using real-space refinement in coot with occasional jelly-body, and at later stages later reciprocal-space, refinement with Refmac to maintain sensible geometry. Near the end, phenix.real_space_refine (*Afonine et al., 2013*) was used to tightly regulate geometry and correct ramachandran and rotamer outliers. After manually checking the corrected model, Phenix was run a final time in ADP mode to assign atom B-factors and generate the final model statistics (*Table 1*). Coordinates were deposited with PDB ID code 5LD2.

## Acknowledgements

We thank Alistair Siebert, Daniel Clare and Sonja Welsch at the EBIC EM facility at Diamond for help with EM data collection, Garib Murshudov for providing refinement scripts, Wojtek Krajewski and Xin Fu for providing plasmid pETduet-His$_6$-TEVsite-recB$^{D1080A}$. The work was funded by Cancer Research UK, the Medical Research Council and the Wellcome Trust.

## Additional information

### Funding

| Funder | Grant reference number | Author |
| --- | --- | --- |
| Cancer Research UK | A21608 | Dale B Wigley |
| Wellcome Trust | WT095519MA | Dale B Wigley |
| Medical Research Council | MR/N009258/1 | Dale B Wigley |

The funders had no role in study design, data collection and interpretation, or the decision to submit the work for publication.

### Author contributions

MW, Conception and design, Acquisition of data, Analysis and interpretation of data, Drafting or revising the article; YC, Acquisition of data, Analysis and interpretation of data, Drafting or revising the article; DBW, Conception and design, Analysis and interpretation of data, Drafting or revising the article

### Author ORCIDs

Dale B Wigley, http://orcid.org/0000-0002-0786-6726

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
