## [Decision Letter]

Thank you for submitting your article "Mechanism for nuclease regulation in RecBCD" for consideration by *eLife*. Your article has been favorably evaluated by Richard Losick (Senior editor) and three reviewers, one of whom is a member of our Board of Reviewing Editors. The following individual involved in review of your submission has agreed to reveal his identity: Stewart Shuman (Reviewer #2).

The reviewers have discussed the reviews with one another and the Reviewing Editor has drafted this decision to help you prepare a revised submission.

In this work, Wigley and colleagues extend their highly informative structural studies of bacterial helicase-nucleases by reporting the cryo-EM structure of RecBCD engaged with a forked DNA substrate in which the 5' single strand tail is long enough to traverse the RecD subunit. This new structure reveals conformational changes underlying the requirement for RecD to activate the nuclease activity inherent to the C-terminal domain of the RecB subunit. This entails the movement of a nuclease-occluding protein helix observed in an earlier crystal structure of RecBCD. The EM structure highlights a key role for an SH3 domain in the RecD subunit in this transition, whereby SH3 (which was disordered in the RecBCD initiation complex crystal structure) interacts with the 5' ssDNA and RecB. This is an important addition to our specific understanding of RecBCD function and its regulation, and more generally an excellent example of the sophisticated internal dynamics of a DNA-processing machine.

I summarize below the relatively minor comments that you need to take into account in your revision. Also note that *eLife* expects papers to refer to the organism system in either the title or Abstract so I would propose you add an appropriate statement in the Abstract stating that is *E. coli* RecBCD that is under study.

1) The simulations in the videos are very informative, but please check them carefully since one reviewer had the opinion that the Video 2 file was corrupted, although I was not aware of a problem. Could the nuclease active site be indicated in the videos, such as by overlaying one of the metal ions?

2) In our opinion, the figures could be improved, not least because the difficulty in distinguishing colors and relating one figure to another. In Figure 1 the green (1A) domain is either hidden or difficult to distinguish from the yellow.

From the way the figures are presented, it is difficult to understand the conformational changes that lead from the binding of the RecD-SH3 domain to the 5'-tail, to the exposure of the nuclease active site. This is in part due to the changes in orientation used for the close-up views (such as in Figure 5), which are difficult to relate back to the starting view (Figure 1). The videos show these changes very well, but many readers will just look at the figures. I am not sure what the best solution is, but one could consider having two full structures shown side by side for the two complexes (present structure and previous "pre-initiation" structure), in the orientation of Figure 1. It would be helpful if all of the key parts that participate in the conformational changes (nuclease active site, RecB linker, etc.) could be labeled or indicated on each figure. At a minimum, it would be helpful if the nuclease active site could be indicated on Figure 1 – one should be able to see that it is buried in the pre-initiation complex and exposed in the present structure, which is the main point of the paper, but is not clearly shown anywhere. It would be better if the close-up views that show the detailed interactions, could be in a similar orientation as Figure 1, if at all possible, ideally on the same panel with boxes on the full structure indicating each expanded region. I realize that this is difficult however, and I like how the overlays with the gray structure show the actual domain movements.

Figure 5 – please indicate the location of the nuclease active site. Furthermore, it would be nice if the domain structure of each subunit could be defined in a linear sequence cartoon (color coded like the structural figures). With so many different domains and subdomains, it would be nice to have a legend to show where the different domains are in the linear sequence.

Since much of the text and figures (Figure 2 and Figure 3) have to do with the ATP sites. It would be helpful if the previous structures that were determined were described more fully (such as in the introduction), with regard to the occupancy of the two ATP sites. It wasn't clear if the occupancy of the ATP sites, and the conformational changes around them, had anything to do with the nuclease activation, or if this was just a side note of the paper. Since the main point of the paper has to do with nuclease activation, are the parts about the ATP sites a bit off-track, although I understand that they are still informative? If there is in fact a connection between the ATP sites and nuclease activation, it should be made clearly.

The authors might also consider the value of introducing a cartoon of the pathway from DNA loading to nuclease action, chi recognition etc. that takes into account the structural work here and previously; this could be very useful for the general reader.

3) Correct and check for typos (some of which include 'These' to 'This' [end of subsection “Structure determination by cryoEM”]; insert 'the' before 'overall' [subsection “Mechanism of nuclease activation”, second paragraph]; 'collected' not coleted' [subsection “Cryo-electron microscopy grid preparation and data collection”, last paragraph]; no title on He et al. ref in citations. Also the authors should carefully examine their revised manuscript for accuracy, particularly the Methods section.

4) In the Abstract and first paragraph of the Introduction, it would be appropriate to cite AdnAB (the mycobacterial helicase-nuclease) along with AddAB and RecBCD as exemplary bacterial DSB-processing machines.

5) In the second paragraph of the Introduction. The authors cite both RecCD and AddAB as antagonists of bacteriophage infection, and then cite several cases in which phages encode proteins that neutralize RecBCD. Yet, they cite no example of AddAB having an antiviral effect. is there such evidence? If not, then AddAB ought to be deleted from the sentence.

6) In the second paragraph of the subsection “Mechanism of nuclease activation”, please find better phrasing than "activates activity".

---

## [Author Response]

*I summarize below the relatively minor comments that you need to take into account in your revision. Also note that eLife expects papers to refer to the organism system in either the title or Abstract so I would propose you add an appropriate statement in the Abstract stating that is E. coli RecBCD that is under study.*

*1) The simulations in the videos are very informative, but please check them carefully since one reviewer had the opinion that the Video 2 file was corrupted, although I was not aware of a problem. Could the nuclease active site be indicated in the videos, such as by overlaying one of the metal ions?*

We have uploaded the videos again in case of any issues experienced by one of the reviewers.

2) In our opinion, the figures could be improved, not least because the difficulty in distinguishing colors and relating one figure to another. In Figure 1 the green (1A) domain is either hidden or difficult to distinguish from the yellow.

*From the way the figures are presented, it is difficult to understand the conformational changes that lead from the binding of the RecD-SH3 domain to the 5'-tail, to the exposure of the nuclease active site. This is in part due to the changes in orientation used for the close-up views (such as in Figure 5), which are difficult to relate back to the starting view (Figure 1). The videos show these changes very well, but many readers will just look at the figures. I am not sure what the best solution is, but one could consider having two full structures shown side by side for the two complexes (present structure and previous "pre-initiation" structure), in the orientation of Figure 1. It would be helpful if all of the key parts that participate in the conformational changes (nuclease active site, RecB linker, etc.) could be labeled or indicated on each figure. At a minimum, it would be helpful if the nuclease active site could be indicated on Figure 1 – one should be able to see that it is buried in the pre-initiation complex and exposed in the present structure, which is the main point of the paper, but is not clearly shown anywhere. It would be better if the close-up views that show the detailed interactions, could be in a similar orientation as Figure 1, if at all possible, ideally on the same panel with boxes on the full structure indicating each expanded region. I realize that this is difficult however, and I like how the overlays with the gray structure show the actual domain movements.*

*Figure 5 – please indicate the location of the nuclease active site. Furthermore, it would be nice if the domain structure of each subunit could be defined in a linear sequence cartoon (color coded like the structural figures). With so many different domains and subdomains, it would be nice to have a legend to show where the different domains are in the linear sequence.*

*Since much of the text and figures (Figures 2 and 3) have to do with the ATP sites. It would be helpful if the previous structures that were determined were described more fully (such as in the introduction), with regard to the occupancy of the two ATP sites. It wasn't clear if the occupancy of the ATP sites, and the conformational changes around them, had anything to do with the nuclease activation, or if this was just a side note of the paper. Since the main point of the paper has to do with nuclease activation, are the parts about the ATP sites a bit off-track, although I understand that they are still informative? If there is in fact a connection between the ATP sites and nuclease activation, it should be made clearly.*

*The authors might also consider the value of introducing a cartoon of the pathway from DNA loading to nuclease action, chi recognition etc. that takes into account the structural work here and previously; this could be very useful for the general reader.*

We have made multiple improvements to the figures (and figure legends) to address the (very valid) concerns raised. We hope these greatly improve the explanation of the conformational changes and we feel these changes have improved the manuscript immeasurably!

Figure 1 – We have made extensive changes to the colour schemes to help distinguish the relevant parts of the structure. We now include a new panel to serve as a “road map” with the standard view of the structure coloured by relevant domains to help orient the reader in the subsequent figures. We also include insets on subsequent zoomed in figures to help readers to locate the detailed regions in the overall structure. As suggested, we have also included a coloured bar to represent the linear sequence of the proteins. Several panels are now moved to relevant sections of the supplementary material.

Figure 2 – RecB specific information now moved to Figure 1—figure supplement 2. The new Figure 2 now shows figures relating to the binding of the ssDNA tail to the RecD subunit. It includes an inset to help readers locate the site within the complex.

Figure 3 – ATP-binding site information has been moved to Figure 2—figure supplement 1. Figure 3 now describes the SH3 domain interactions (formerly Figure 4) but now has an inset to orient readers. The electron density figure has been moved to the Supplementary.

Figure 4 – This has become Figure 3. The new Figure 4 replaces the old Figure 5. An inset orients reader and colour schemes/views relate to Figure 1. We hope this clarifies the conformational changes that progress from the initiation complex to nuclease activation.

Figure 5 – This has become Figure 4. In its place, as requested, we now include a cartoon to illustrate the positions of the various structural intermediates on the RecBCD pathway and have included the PDB ID codes for these structures.

Figure 6 – now moved to Supplementary.

In addition, we now include several Supplementary Figures:

Figure 1—figure supplement 1 – The EM data statistics and structure validation have been collated.

Figure 1—figure supplement 2 – Aspects of the electron density to illustrate the quality of different regions of the final map.

Figure 2—figure supplement 1 – most of the old Figure 2 has now been shifted here to improve the flow of the main text.

Figure 2—figure supplement 2 – A modified version of the old Figure 3 is included here. The ATP-binding site issues do not relate to the conformational changes we observe so as suggested we have extracted this from the main text but for completeness we wish to keep it within the Supplementary material.

Figure 4—figure supplement 1 – Two panels of the old Figure 5 have been moved here to show the electron density for the relocated linker peptide and nuclease blocking helix.

*3) Correct and check for typos (some of which include 'These' to 'This' [end of subsection “Structure determination by cryoEM”]; insert 'the' before 'overall' [subsection “Mechanism of nuclease activation”, second paragraph]; 'collected' not coleted' [subsection “Cryo-electron microscopy grid preparation and data collection”, last paragraph]; no title on He et al. ref in citations. Also the authors should carefully examine their revised manuscript for accuracy, particularly the Methods section.*

All of the typos have been corrected and the Methods section has been checked thoroughly for accuracy.

*4) In the Abstract and first paragraph of the Introduction, it would be appropriate to cite AdnAB (the mycobacterial helicase-nuclease) along with AddAB and RecBCD as exemplary bacterial DSB-processing machines.*

AdnAB has now been cited in the Introductory section.

*5) In the second paragraph of the Introduction. The authors cite both RecCD and AddAB as antagonists of bacteriophage infection, and then cite several cases in which phages encode proteins that neutralize RecBCD. Yet, they cite no example of AddAB having an antiviral effect. is there such evidence? If not, then AddAB ought to be deleted from the sentence.*

The effect of AddAB was presumed on the basis of RecBCD examples. While it is likely correct, the reviewers were correct to point out there are as yet no documented examples, as far as we are aware, so we have now deleted reference to AddAB.

*6) In the second paragraph of the subsection “Mechanism of nuclease activation”, please find better phrasing than "activates activity".*

The phrase has been altered to “[…]and allows access to DNA.”